# Mouse Spinal Cord Vascular Transcriptome Analysis Identifies CD9 and MYLIP as Injury-Induced Players

**DOI:** 10.3390/ijms24076433

**Published:** 2023-03-29

**Authors:** Isaura Martins, Dalila Neves-Silva, Mariana Ascensão-Ferreira, Ana Filipa Dias, Daniel Ribeiro, Ana Filipa Isidro, Raquel Quitéria, Diogo Paramos-de-Carvalho, Nuno L. Barbosa-Morais, Leonor Saúde

**Affiliations:** 1Instituto de Medicina Molecular João Lobo Antunes, Faculdade de Medicina da Universidade de Lisboa, 1649-028 Lisboa, Portugal; 2Instituto de Medicina Molecular João Lobo Antunes e Instituto de Histologia e Biologia do Desenvolvimento, Faculdade de Medicina da Universidade de Lisboa, 1649-028 Lisboa, Portugal

**Keywords:** spinal cord injury, perivascular cells, blood spinal cord barrier, endothelial cells, pericytes, transcriptome

## Abstract

Traumatic spinal cord injury (SCI) initiates a cascade of cellular events, culminating in irreversible tissue loss and neuroinflammation. After the trauma, the blood vessels are destroyed. The blood-spinal cord barrier (BSCB), a physical barrier between the blood and spinal cord parenchyma, is disrupted, facilitating the infiltration of immune cells, and contributing to a toxic spinal microenvironment, affecting axonal regeneration. Understanding how the vascular constituents of the BSCB respond to injury is crucial to prevent BSCB impairment and to improve spinal cord repair. Here, we focus our attention on the vascular transcriptome at 3- and 7-days post-injury (dpi), during which BSCB is abnormally leaky, to identify potential molecular players that are injury-specific. Using the mouse contusion model, we identified *Cd9* and *Mylip* genes as differentially expressed at 3 and 7 dpi. CD9 and MYLIP expression were injury-induced on vascular cells, endothelial cells and pericytes, at the injury epicentre at 7 dpi, with a spatial expression predominantly at the caudal region of the lesion. These results establish CD9 and MYLIP as two new potential players after SCI, and future studies targeting their expression might bring promising results for spinal cord repair.

## 1. Introduction

After a mammalian traumatic spinal cord injury (SCI), there is an induced synaptic loss and neuronal cell death. It prompts a widespread deposition of cellular debris that triggers gliosis and neuroinflammation [1,2]. Reactive gliosis gives rise to a mature astrocytic border, within which resides a fibrotic scar, representing a significant physical barrier to axonal regeneration [3,4]. Once activated, microglia, neutrophils, macrophages, and lymphocytes initiate a secondary damage response by releasing chemokines and cytokines, contributing to a strong inflammatory environment and constraining the regenerative capacity [5]. In addition, the actual mechanical force of the trauma disrupts the spinal cord vasculature, damaging all the components of the neurovascular unit (NVU) of the blood-spinal cord barrier (BSCB), which can remain compromised [6].

The NVU is a crucial structure composed of endothelial cells (ECs), pericytes, astrocytic endfeet and neurons responsible for the normal homoeostasis of the spinal parenchyma, protecting the cord during trauma and infection [7,8]. However, this system is compromised after injury, and BSCB is damaged. This scenario favours the increased circulation of immune cells and pro-inflammatory mediators, contributing to systemic inflammation [9], and induces ischemia, leading to greater tissue loss [10]. The degree of immune infiltrates into the cord has been associated with the status of BSCB in SCI [11]. In addition, pericytes from the NVU can detach from the vascular wall and contribute to scar formation after injury [12], demonstrating another important role of the components of the BSCB in SCI.

Although there is an angiogenic response to the injury, the new blood vessels formed in the lesion area are usually leaky with an impaired BSCB, further contributing to secondary damage [13]. Furthermore, abnormal permeability of BSCB is even more apparent during the initial process of angiogenesis, which proceeds during the sub-acute phase (3–7 days post-injury (dpi)) in the mouse model [14], further contributing to a hostile microenvironment that limits axonal regeneration and repair. Thus, acting during this period, when the BSCB is even more prone to impairment, might prevent irreversible damage and diminish the secondary injury.

The promotion of BSCB integrity and blood vessel vascularization/remodelling after SCI are crucial for spinal cord repair [14,15,16,17]. Nevertheless, the numerous therapeutic interventions based on revascularization and targeting BSCB permeability have achieved limited therapeutic effects [16,17]. It demonstrates that there is still the need to better understand the instilled molecular changes on BSCB after injury, particularly at early stages, to identify potential players that could promote both vascularization and BSCB integrity and therefore stimulate regeneration.

In this study, we explored the transcriptome of spinal cord vascular cells at 0, 3 and 7 dpi, a period in which the process of angiogenesis affects BSCB integrity, to characterize the molecular basis of the response to injury in a mouse contusion SCI model. We identified two genes, *Cd9* and *Mylip*, whose expression is injury-induced. CD9 and MYLIP proteins were found in association with vascular and/or perivascular cells (ECs and pericytes) and accumulate at the injury periphery, predominantly at the caudal region. Therapeutic strategies in future studies aimed at exploring their potential in spinal cord repair by targeting CD9 and MYLIP during the subacute phase of the injury may hold promising results for BSCB control and functional recovery.

## 2. Results

### 2.1. Validation of Vascular Identity of Sorted Cells from Spinal Cord Samples

To explore the vascular transcriptome, spinal cords from sham and injured animals (0, 3 and 7 dpi) were dissociated for single-cell analysis by Fluorescence Activated Cell Sorting (FACS) using the vascular marker CD31. Due to the complex microenvironment of the BSCB, rich in cell contacts and adhesion molecules, it can be hard to dissociate single cells. Therefore, we used CIBERSORTx [18], a “digital cytometry” tool that estimates relative cell type abundances from bulk tissue transcriptomes, to assess the purity of our samples at 3 and 7 dpi (n = 2–3) and their composition in different cell populations. For this, we resorted to gene expression signatures of the major cell types in the mouse spinal cord, which are vascular, immune, glial and neuronal populations as defined by the Brain Mouse Atlas [19]. Considering these main cell populations, we estimated an average of 67% of mRNA in the analyzed samples to be of vascular and perivascular origin, including ECs, pericytes and perivascular macrophages (Figure 1). This estimate is concordant with the method applied to sort vascular cells. The second most abundant cell type appeared to be of immune origin (average of 16%) (Figure 1). It is not surprising, given the close interaction of immune cells and the BSCB, which can be sorted following the incomplete dissociation of the vascular cells. With this validation, we were confident that the studied transcriptomes were mostly of vascular origin.

### 2.2. Identification of Differentially Expressed Genes in Vascular Cells after Spinal Cord Injury

Differential gene expression was analyzed for the subset of high-confidence genes considered, i.e., genes whose expression was detected and measured in all samples (see Materials & Methods), to identify GE alterations that are specific to injured samples at 3 and 7 dpi, respectively. 13 genes were identified as significantly differentially expressed (B statistic > 0) at 3 dpi (when compared with 0 dpi) specifically in SCI samples (Figure 2a, Table 1). 5 of these 13 genes were identified as also significantly differentially expressed at 7 dpi (when compared with 0 dpi) specifically in SCI samples (Figure 2b, Table 1). From those 13 genes specifically altered in injury at 3 dpi, 8 were upregulated (*Mylip, Cd9, Selplg, Map3k1, Slc3a2, Rpr16, Ssr1, H2-DMb2*) and 5 downregulated (*Nek7, Trak2, Rock1, Ppk4, S100β*). At 7 dpi, from those 5 differentially expressed genes, 3 showed upregulation (*Cd9, Mylip, Scl3a2*), and 2 were downregulated (*Nek7* and *Rock1*). Given that they were consistently the two injury-specifically most over-expressed genes at 3 and 7 dpi (Table 1), we selected *Cd9* and *Mylip* as candidate SCI-associated players at the perivascular microenvironment at 3 and 7 dpi.

### 2.3. Spinal Cord Injury Induces an Upregulation of Immune-Related Biological Processes in Vascular Cells

To understand whether the modulation of the 13 genes we identified at 3 dpi was part of a larger transcriptional program orchestrated by the vascular cells upon injury, we performed Gene Set Enrichment Analysis (GSEA) [20] to look for potentially enriched biological processes. Interestingly, we found that most terms positively enriched were immune-related, including those associated with immune regulation, immune effector function, immune response regulating cell surface receptor signalling, leukocyte migration and activation processes (Figure 3).

### 2.4. CD9 and MYLIP mRNA and Protein Levels Increase after Spinal Cord Injury

Given that the differential gene expression data obtained from spinal cord vascular cells showed *Cd9* and *Mylip* as two genes being overexpressed after injury in both time points, we focused our analysis on validating their expression in total spinal cord tissue at 3 and 7 dpi. First, we evaluated their mRNA levels by qPCR. Both *Cd9* (Figure 4a) and *Mylip* (Figure 4b) were significantly increased after injury when compared with sham controls at 3 and 7 dpi, as detected in the transcriptome analysis.

To assess if the increase of mRNA levels of *Cd9* and *Mylip* was translated into increased protein levels, we performed a Western blot using 6 mm of spinal cord tissue spanning from the injury or sham epicentre. No changes were found between sham and SCI samples at 3 dpi for both CD9 (Figure 5a) and MYLIP (Figure 5b). However, at 7 dpi, CD9 and MYLIP protein levels were significantly increased compared to controls (Figure 5a, b). Taken together, these results are consistent with the vascular transcriptome data and suggest an increase in protein translation of CD9 and MYLIP at 7 dpi.

### 2.5. CD9 and MYLIP Expression Is Injury-Induced Predominantly at the Caudal Lesion Site

We determined the spatial localization of CD9 and MYLIP expression at 7 dpi in longitudinal cryosections of injured and sham spinal cords, a time-point in which we observed an increase in protein levels for both proteins. In sham-injured animals, no expression of both CD9 (Figure 6A–C) and MYLIP (Figure 7A–C) was observed in all animals analyzed (n = 3). However, after injury, CD9 and MYLIP were detected at the injury site with more prominent expression in the caudal region of the lesion in all animals analyzed (n = 3) (Figure 6D–F’ and Figure 7D–F’).

### 2.6. CD9 Expression Is Detected in Close Proximity with Endothelial and Pericytic Markers

To explore CD9 expression after injury in situ, we further investigated its expression in association with endothelial and pericytic markers. Using CD31 as a marker for ECs, and αSMA for pericytes (see Materials & Methods, Appendix A), we were able to identify that when observed, CD9 is in close proximity with both vascular and perivascular markers (both ECs and pericytes) (Figure 8).

### 2.7. MYLIP Expression Is Detected in Association with Pericytic Markers, including in Pericytes Detached from Blood Vessels

To investigate the injury-induced expression of MYLIP in the spinal cord, CD31 and CD13 were used (see Materials & Methods, Appendix A) to assess both endothelial and pericytic populations, respectively. We observed that, in contrast to CD9, MYLIP expression was only observed in close proximity to the marker CD13 (Figure 9). In addition, MYLIP is also detected in pericytes that are not attached to blood vessels (Figure 10).

## 3. Discussion

In mammals, SCI leads to irreversible tissue loss, characterized by a chronic state of neuroinflammation. The complexity of the disorder involves two waves of injury: the first associated with the physical damage to the cord caused by the trauma; and a second injury produced by a series of molecular and cellular events that perpetuate tissue dysfunction, inhibiting axonal regeneration and preventing functional recovery [21].

One of the main structures that are destroyed by the mechanical force of the trauma is the vasculature [15], which in turn is one of the key players in disseminating the second injury and propagating neuroinflammation [9,10,13]. Once broken down, the blood vessels become dysfunctional, and the cellular complex they are part of—the BSCB—becomes leaky, allowing the infiltration of immune cells and inhibiting blood supply to the spinal cord. Therefore, several studies in the past years have focused on therapeutic strategies to promote the regeneration of the vasculature and restore BSCB integrity to create an optimal microenvironment for neuronal repair [14,17]. Unfortunately, little to no recovery has been attained, demonstrating the complexity of the vascular response after trauma and the importance of other unknown molecular players.

This study used the contusion mouse model of SCI to explore the vasculature response after a traumatic injury. We focused our attention during the subacute injury phase, between 3 to 7 dpi, when the abnormal permeability of BSCB is even more apparent, which precedes the settlement of scar formation. We hypothesized that, by resorting to RNA sequencing analysis of the vascular population at the spinal cord injury epicentre, we might have a sight of key vascular players that could be involved in BSCB integrity and could be targeted in future studies.

To understand the vascular response, we used CD31, a specific marker of vascular differentiation, to isolate spinal cord vascular cells from the injury or sham epicentre at 0, 3 and 7 dpi. The FACS-cells were sequenced, and CIBERSORTx deconvolution [18] was used to estimate their cellular composition using inferred gene expression signatures for each of the significant cellular populations (vascular, immune, glial and neuronal) present in the spinal cord. This analysis allowed us to estimate the purity of the isolated cells.

We observed an average vascular purity of 67% in the sorted samples (Figure 1). However, knowing the complexity of the vasculature, with high cell-to-cell contacts and adhesion molecules, we cannot exclude that single-cell dissociation might not have been completely accomplished and some cell duplets might have been sorted, in particular of cells that are part of the BSCB, such as pericytes and perivascular macrophages. Our CIBERSORTx analysis identifies the vascular population of ECs, pericytes and perivascular macrophages. Our second main cellular population were immune cells (Figure 1), particularly high in injured samples (SCI 3 dpi: 35.4%; SCI 7 dpi: 17.0%) when compared with controls (sham 3 dpi: 3.3%; sham 7 dpi: 5.9%) (Figure 1). As it is known, particularly in the case of injured spinal cords, immune cell infiltrates can considerably affect the sorting of vascular cells [22]. Furthermore, although CD31 is fairly specific for vascular cells, some reactivity may also be seen in macrophages and other immune cells [23]. Therefore, we must consider the 15% contribution of immune cells in our analysis and not exclude the input of immune cells in BSCB integrity and dysfunction.

With a purity of approximately 70% in our FACS-based isolation, we next analyzed the expression dynamics for the subset of high-confidence genes (i.e., genes with detectable expression in all samples). We identified 13 genes specifically differentially expressed in SCI samples at 3 dpi (Figure 2a), and 5 genes within these 13 were identified as also specifically differentially expressed in SCI samples at 7 dpi (Figure 2b). To further investigate the changes in the transcriptome after SCI in the vasculature, we used GSEA at 3 dpi, the time-point in which we see more differentially expressed genes (Figure 3). GSEA revealed that after an SCI, there is an upregulation of immune-related gene ontologies in our FACS-isolated vascular cells, concomitant with the subacute phase in which the cells were sorted and with our “digital cytometry” analysis (16% of the immune population). In addition, we observed the activation of processes associated with regulating immune effector, cell surface receptor signalling, myeloid leukocyte migration and lymphocyte activation (Figure 3). As known, all the cellular components of the NVU play an important role in the immune response during homeostasis and injury. Therefore, it is expected that, particularly after SCI, these cells activate these same pathways in the first instance to respond to the damage but later in a chronic and disruptive manner that contributes to neuroinflammation. In contrast, we observed a downregulation of processes associated with synaptic and cell-to-cell signalling, and axonal transport correlated with the aggressive microenvironment of the spinal tissue as the result of the injury (Figure 3).

We focused our attention on the 2 top genes identified as overexpressed in both time points with the highest B statistic: *Cd9*, which encodes an endothelial tetraspanin protein that, although expressed in a variety of cell types, is particularly high in ECs [24,25], where it plays a crucial role in the transendothelial migration of leukocytes [24,26,27]; and *Mylip*, which encodes a novel ERM-like protein that interacts with myosin regulatory light chain, that inhibits neurite outgrowth [28] and shown to be overexpressed in pericytes during tumour progression [29].

To further validate these results, CD9 and MYLIP mRNA and protein levels were assessed at 3 and 7 dpi. mRNA levels were significantly increased for both *Cd9* and *Mylip* (Figure 4). However, this increase was only translated into augmented protein levels at 7 dpi for both proteins (Figure 5). It proposes that, although there is a significant increase in gene expression at 3 dpi, the effector protein may only have an important role at 7 dpi after injury. Considering that 7 dpi appears to be an important time-point where CD9 and MYLIP protein levels were significantly increased, we next pursue the in-situ validation at 7 dpi.

Our histological analysis revealed that both CD9 and MYLIP are injury-induced, as no expression was observed in sham animals (Figure 6A–C and Figure 7A–C). Furthermore, CD9 and MYLIP expressions were detected at the injury site with a caudal predominance (Figure 6D–F’ and Figure 7D–F’). As expected, when detected, both CD9 and MYLIP exhibited perivascular expression. CD9 was associated with the endothelial marker CD31 and the pericytic marker αSMA (Figure 8). As pericytes and ECs can share the same membrane in specific vessel locations, denominated as a peg–socket pockets [30], it is unsurprising that CD9 expression might be shared between ECs and pericytes at different levels in the vasculature. CD9 expression has already been described as a crosstalk signalling peptide between ECs and pericytes, particularly in pocket regions where both cells share the intramembrane space [31,32], reinforcing the importance of our results. In addition, other studies have already associated CD9 with SCI, both in proteomic data of contusion rat models at later time points (8 weeks) [33] and in mouse cervical injury [34]. Nevertheless, although this supports CD9 as injury-induced, this is the first time CD9 expression is associated with vascular and perivascular populations after SCI.

On the other hand, MYLIP expression was revealed to be only associated with pericytes due to its proximity to the pericytic marker CD13 (Figure 9 and Figure 10). MYLIP was also present in a small population of pericytes dissociated from blood vessels (Figure 10) and close to the lesion epicentre, suggesting that these pericytes might contribute to scar formation. MYLIP expression has already been identified in pericytes in other contexts, such as in transcriptomic data of tumours [29] and single-cell profiling of lung tissue [35]. However, to our knowledge, no study has ever shown MYLIP expression associated with pericytes in the context of an SCI.

Pericytes regulate the capillary tone and blood flow in the spinal cord below the site of the lesion [36], suggesting that there are pericyte-mediated constriction mechanisms that decrease spinal blood flow below the lesion that, if identified, can contribute to a decrease in hypoxia and improve myelination and regrowth. We hypothesize that both CD9 and MYLIP might play a role in such mechanisms but also have a crucial part in the inflammatory response, although further research should be done in the near future. CD9, in particular, already has a described role in immune control [24,26,27] but can also dynamically interact with other transmembrane and cytoplasmatic proteins and therefore have a multitude of biological functions, such as affecting the activity of metalloproteinases, cytokines and chemokines, among others [25]. In addition, MYLIP could regulate the integrity of the BSCB as it plays a key role in maintaining cellular morphology, modulation of cell motility, remodelling of cytoskeletal proteins, and adhesion of cells with extracellular matrix (ECM) and other cells through ICAM-1 and integrins [37].

Here, we identify for the first time CD9 and MYLIP as new vascular players in SCI. However, further research still needs to be done to reveal their function and associated mechanisms of action. Nevertheless, we anticipate that forthcoming strategies targeting CD9 and/or MYLIP may hold promising results for BSCB integrity by decreasing inflammatory response and promoting a more permissive regenerative spinal microenvironment.

## 4. Materials and Methods

### 4.1. Animals

All surgical and postoperative care procedures were performed in accordance with the Federation of European Laboratory Animal Science Associations (FELASA) guidelines and were approved by Instituto de Medicina Molecular—João Lobo Antunes, Faculdade de Medicina, Universidade de Lisboa, Portugal (iMM) and followed the Portuguese Animal Ethics Committee (DGAV) regulations. Adult (8–9 weeks old) female C57BL/6J mice (*Mus musculus*) were purchased from Charles River Laboratory and housed in the animal facilities of iMM. Animals were housed under conventional conditions on a 12 h light-dark cycle with *ad libitum* access to food and water.

### 4.2. Spinal Cord Injury and Post-Operative Care

After two weeks of handling and acclimatization, body weight was assessed to ensure ideal weight (18–20 g), and animals were assigned to spinal cord injury. C57BL/6J mice (9–11 weeks old) were anesthetized using a cocktail of ketamine (120 mg/kg) and xylazine (16 mg/kg) administered by intraperitoneal (ip) injection. For spinal contusion injuries, a laminectomy of the ninth thoracic vertebra (T9), identified based on anatomical landmarks, was first performed, followed by a moderate (75 kdyne) contusion using the Infinite Horizons Impactor (Precision Systems and Instrumentation, LLC.). After SCI, the muscle and skin were closed with 4.0 polyglycolic absorbable sutures (Safil, G1048213). In control uninjured mice (sham), the wound was closed and sutured after the T9 laminectomy, and the spinal cord was not touched. Animals were injected with saline (0.5 mL) subcutaneously (sq) and then placed into warmed cages until they recovered from anaesthesia and for the following recovery period (3 days). To prevent dehydration, mice were supplemented with daily saline (0.5 mL, sq) for the first 5 dpi. Bladders were manually voided twice daily for the duration of experiments.

### 4.3. Single-Cell Preparation for FACS

Animals were sacrificed at 0, 3 and 7 dpi, and the spinal cords were harvested for Fluorescence Activated Cell Sorting (FACS). Approximately 6 mm spanning the injury/sham epicentre of the manipulated experimental and control spinal cords were collected, and for each condition, 3/4 biological replicates were used. The harvested spinal cord samples were homogenized according to a spinal cord-specific and FACS-compatible protocol optimized in our laboratory. Briefly, each spinal cord was dissected in DMEM and then transferred to a specific digestion mix (0.01% CaCl, 200 U/mL Collagenase I (Sigma 680 U/mg), 0.000125% of 2% DNAseI in DMEM (GIBCO), for 30 min at 37 °C to allow digestion of the spinal cord tissue. 22% BSA was added in a 1:1 ratio to allow the separation between myelin and the vascular tubes, followed by centrifugation at 1360 g for 10 min at 4 °C. After the removal of myelin, cold EC medium (DMEM + 10% FBS) was added and the suspension of cells was filtered through a 70 µm filter to remove undigested cell clumps and separate single cells. Additional steps were added to eliminate blood cells by adding ACK lysing buffer (GIBCO) followed by a wash with a FACSmax Cell Dissociation Solution (Amsbio). The cell suspension was incubated with a rat anti-mouse CD31-PE (BD Biosciences 1:50, 561073) on ice and in the dark for 45 min, followed by a wash of FACS solution and spin at 400 g for 5 min. Cells were resuspended in FACS medium, filtered, and 7AAD (Miltenyi Biotec, 130-111-568) added. All samples were passed through a FACSAriaIII Cell Sorter to separate the specific vascular cells from non-endothelial and cell death fractions. The EC fractions were collected in RLT-plus buffer (Qiagen, 1053393) and stored at −80 °C until RNA extraction was performed the following day.

### 4.4. Preparation of cDNA Library and RNA-Seq

Cells in suspension were collected in 2.5 µL of Buffer RLT Plus (Qiagen, 1053393), and an mRNA library was prepared at IGC Genomics Unit using SMART-Seq [38]. Illumina libraries were generated with the Nextera-based protocol, and library quality was assessed in Fragment Analyzer before sequencing. Sequencing was performed in NextSeq 500 Sequencer (Illumina) at the IGC Genomics facility using SE75bp and 30 million reads per library. Sequencing data were demultiplexed and converted to FASTQ format using bcl2fastq v2.19.1.403 (Illumina).

### 4.5. Pre-Processing of RNA-Seq

Provided FASTQ files with RNA-seq data from the six studied conditions (SCI and sham samples for each time-point considered: 0, 3 and 7 days post-injury) were checked for the overall quality of sequencing reads using the tool fastqc (https://www.bioinformatics.babraham.ac.uk/projects/fastqc/ accessed on 28 August 2018) and information was compiled using MultiQC on 28 August 2018 [39]. Despite differences in the total number of sequenced reads per sample (addressed in the following steps), all samples had enough quality. Vast-tools (Vertebrate Alternative Splicing and Transcription Tools) version 2 [40] was used for alignment and quantification of gene expression, considering the VastDB [41] gene annotation for mouse genome assembly mm9. The total number of aligned reads and profiled genes per sample were inspected, and only samples with more than 7 million read counts were considered for further analysis (Appendix A). Furthermore, sample *0dpisham3b_S_Run7_12* was not considered due to an abnormally low number of profiled genes compared to all other considered samples, suggesting a library of very low complexity. Thus, from the 24 original samples, only 15 were further considered. The RNA-seq data have been deposited in the Sequence Read Archive (accession number provided upon request).

### 4.6. Gene Expression Quantification

Using the vast-tools pipeline, gene expression quantification is performed by aligning RNA-seq reads against a reference transcriptome and counting those mapping to each given gene. Raw read counts for 22,667 genes were obtained by vast tools. To consider only the genes for which expression was detected in all samples, only genes with at least 1 read count for all samples and with read count variance different from 0 were considered (1266 genes). To prepare count data for differential gene expression analysis, normalization factors to scale raw library sizes were obtained using the function calcNormfactors from package edgeR [42] after converting the filtered count matrix into a *DGEList* object. The voom method with quantile normalization [43] was then applied to count data to obtain gene expression estimates in log_2_-counts per million (logCPM).

### 4.7. Derivation of Gene Expression Signatures for the Major Spinal Cord Cell Types

We employed CIBERSORTx [18] to infer gene signatures of major cell types present in the mouse spinal cord. The tutorials provided at the CIBERSORTx portal (https://cibersortx.stanford.edu/, accessed on 23 June 2022) were used as a basis for our analysis. The Zeisel dataset [19] was used to extract single-cell gene expression data from spinal cord mouse tissue (l5_all.loom file from mousebrain.org). Cells in the dataset are hierarchically classified according to Taxonomy levels. The TaxonomyRank1 is the top level we used to create the signature matrix. A total of 50 cells of each taxon were sampled and renamed according to their taxon (expression_single_cell50_TaxonomyRank1.txt). Signatures were generated with the default parameter except for the following: Min. Expression: 1; Replicates: 20; Sampling: 0.

### 4.8. Estimation of Cellular Composition on FACS-Sorted Samples

We used CIBERSORTx deconvolution to estimate the cellular composition of FACS-sorted samples from bulk RNA sequencing. To build the mixture file containing bulk RNA-seq expression profile, we built a count matrix containing only the high-confidence genes described in Section 4.6 (mouse_per_sample_high_conf_genes.txt) at the analyzed time points. We ran the CIBERSORTx deconvolution algorithm with default parameters except for the following: Batch correction mode: S-mode; Single cell reference matrix file used for S-mode batch correction: expression_single_cell50_TaxonomyRank1.txt; Permutations: 500.

### 4.9. Differential Gene Expression Analysis

Differential gene expression analysis was performed with linear models using the limma R package [43]. The expression of each gene was fitted with a model considering as baseline the average expression level of that gene across samples (centred design) and calculating the increment in expression for each of the following contrasts (coefficients in the model):$$GE=\beta_{baseline}+\beta_{3\mathrm{dpi}}\cdot 3\;\mathrm{dpi}+\beta_{7\mathrm{dpi}}\cdot 7\;\mathrm{dpi}+\beta_{SCI}\cdot SCI+\beta_{3SCI}\cdot 3\;\mathrm{dpi}\cdot SCI+\beta_{7SCI}\cdot 7\;\mathrm{dpi}\cdot SCI$$

(a)3 dpi: GE increment from the 0 dpi average sample to the 3 dpi average sample;(b)7 dpi: GE increment from the 0 dpi average sample to the 7 dpi average sample;(c)Sci: GE increment in the average Sci sample compared to the average Sham sample.(d)Interaction 3 dpi with Sci: GE increment in 3 dpi Sci samples that is not explained by (a) and (c) combined.(e)Interaction 7 dpi with Sci: GE increment in 7 dpi Sci samples that is not explained by (b) and (c) combined.

With such mathematical implementation, the isolated effects of SCI and time (3 or 7 dpi) can be assessed independently from the interaction coefficients β_3SCI_ and β_7SCI_ that, in turn, reflect the GE alterations that are specific to samples *with* the lesion (SCI) at the respective time-points.

Global empirical Bayes statistics are calculated for each gene and each coefficient in the model, allowing for the identification of genes significantly differentially expressed as those with an associated positive log-odds ratio (B statistic > 0) of the gene is differentially expressed and with the magnitude of difference in expression measured in log_2_ fold-change between the two conditions. Using this approach, 13 genes (for the interaction effect between 3 dpi and SCI) and 5 genes (for the interaction effect between 7 dpi and SCI) were identified as significantly differentially expressed.

### 4.10. Gene Set Enrichment Analysis

We used the ‘clusterProfiler’ R package [44] to perform gene set enrichment analysis (GSEA) on the t-statistics of differential gene expression of the high-confidence genes at the analyzed time points. Gene symbols were checked for duplicates, and when found, the symbol with the highest absolute t-statistic value was kept. GO Biological Process ontology genes and terms for *Mus musculus* were retrieved with the package ‘msigdbr’. The GSEA algorithm was run with the default settings, except for the following: pvalueCutoff = Inf, eps = 0, seed = TRUE. We considered terms with adjusted *p*-value < 0.05 to be enriched in our analysis.

### 4.11. Quantitative Real-Time PCR

Total RNA was extracted from mouse spinal cord samples (6 mm of tissue spanning the lesion/sham site) using TRIzol (Invitrogen)/chloroform and purified using the RNA Clean & Concentrator-5 kit (Zymo Research), according to the manufacturer’s instructions. RNA concentration was determined by NanoDrop (Thermo Scientific). cDNA synthesis was performed using the iScript Reverse Transcription Supermix for RT-qPCR (Bio-Rad), according to the manufacturer’s instructions. qPCR was performed using 7500 Fast Real-Time PCR System (Applied Biosystems) and Power SYBR Green PCR Master Mix (Applied Biosystems). For each cDNA sample, three technical replicates were included. Relative mRNA expression was normalized to PPIA mRNA expression using the ΔΔCt method. Primers used for qPCR are listed in Appendix A.

### 4.12. Western Blot

Protein extraction was performed from 6 mm of spinal cord tissue spanning the lesion/sham site of saline-perfused animals. The samples were homogenized in lysis buffer (PBS/1% Triton X-100/protease and phosphatase inhibitors—cOmplete™, EDTA-free Protease Inhibitor Cocktail (Roche, 11873580001). Protein concentration was determined by DC Protein Assay (BioRad, 5000111). Spinal cord extracts were denatured and reduced in 4× Laemmli protein sample buffer (BioRaa, 1610747) supplemented with 10% β-mercaptoethanol and boiled at 95 °C for 10 min. 50 μg of protein per sample was loaded and separated by 4 –15% SDS-PAGE gel (BioRad, 4561084). Proteins were transferred to PVDF membranes (pore size 0.45 μm GE Healthcare, GE10600023) and blocked in 5% bovine albumin serum in TBS/0,1%Tween20 (TBS-T). Membranes were incubated overnight at 4 °C with primary antibodies in a blocking solution. The following day, the membranes were washed in TBS-T and incubated in the respective HRP-conjugated secondary antibody in a blocking solution for 1 h. GAPDH monoclonal antibody was used as a loading control. Membranes were developed using the enhanced chemiluminescence kit Clarity Western ECL Subs (BioRad, 1705060) and visualized at Amersham 680 (GE Healthcare). The intensity of the specific bands was quantified using Image Studio Lite software. Unless described otherwise, all steps were performed at room temperature. More details about the antibodies and membranes are reported in Appendix A.

### 4.13. Immunohistochemistry

To perform immunostaining in sections, the frozen slides were thawed for 30 min. For MYLIP, the slides were washed 3 times with pre-heated PBS in a water bath at 37 °C for 5 min. For CD9 staining, the slides were subjected to antigen retrieval protocol. Afterwards, the slides were incubated for 10 min in a PBS/Triton X-100 and 3 h in blocking. The slides were incubated overnight at 4 °C in a humidified chamber with the specific primary antibodies’ combination in a blocking solution. The following day, a wash in PBS/Triton X-100 solution was performed for 15 min each. The cryosections were incubated with the appropriate secondary antibodies’ combination and 1 mg/mL DAPI (Sigma, D9564) for 2 h, followed by 3 washes in PBS/Triton X-100 for 15 min and 2 times in PBS. The slides were mounted in a Mowiol mounting medium. All steps were performed at room temperature unless described otherwise. Details on antigen retrieval protocol, blocking solutions and antibodies are reported in Appendix A.

### 4.14. Imaging

Longitudinal images of spinal cords were acquired in a Zeiss Cell Observer Spinning Disk (SD) confocal microscope equipped with an Evolve 512 EMCCD camera. Images were acquired with a Plan-Apochromat 20×/0.80 Ph dry objective. DAPI fluorescence was detected using 405 nm for excitation (50 mW nominal output—20% transmission) and a BP 450/50 nm filter, with exposure time set to 150 ms. Alexa Fluor 488 fluorescence was detected using 488 nm for excitation (100 mW nominal output—10% transmission) and a BP 520/35 nm filter, with exposure time set to 100 ms. Alexa Fluor 561 fluorescence was detected using 561 nm for excitation (75 mW nominal output—12% transmission) and a BP 600/50 nm filter, with exposure time set to 100 ms. Alex Fluor 647 fluorescence was detected using 638 nm for excitation (75 mW nominal output—10% transmission) and a BP 690/50 nm filter, with exposure time set to 80 ms. EM Gain was set to 300 for all channels. Z-stacks of the four channels were acquired in tiled regions corresponding to the whole spinal cord tissue (typical acquisition volume ≈ 10 × 1.6 × 0.03 mm), with a Z interval of 0.49 µm and pixel size 0.67 µm. The 3D dataset and maximum intensity projections were stitched in Zeiss ZEN 3.2 (blue edition). Spinal cord close-up images were acquired in a Zeiss LSM 880 laser scanning confocal microscope using an LD C-Apochromat 40×/1.10 water immersion objective. DAPI fluorescence was detected using 405 nm for excitation and a 415–485 nm detection window, with PMT gain set to 500 and offset to 2. Alexa Fluor 488 fluorescence was detected using the 488 nm laser line of an Ar laser for excitation and a 497–541 nm detection window, with GaAsP detector gain set to 500 and offset to 3. Alexa Fluor 561 fluorescence was detected using 561 nm for excitation and a 570–620 nm detection window, with GaAsP detector gain set to 500 and offset to 3. Alexa Fluor 647 fluorescence was detected using 633 nm for excitation and a 680–735 nm detection window, with GaAsP detector gain set to 550 and offset to 3. The pinhole size was set to 1 AU for Alexa Fluor 647, 1.31 AU for Alexa Fluor 561, 1.51 AU for Alexa Fluor 488 and 1.74 AU for DAPI to achieve the same optical slice thickness in all 4 channels. Z-stacks were acquired with Zoom set to 1 (212.55 × 212.55 µm area with 1024 × 1024 frame size—0.21 µm pixel size) with a line average of 2 and 1.02 µs pixel dwell time (unidirectional scan). Maximum intensity projections were performed in Zeiss ZEN 2.3 (black edition), and the image analysis was in the software Fiji. Adobe Illustrator was used for the assembly of figures.

### 4.15. Statistical Analysis

GraphPad Prism 6 was used for data visualization and statistical analysis. qPCR and Western blot data were analyzed using an unpaired Student *t*-test. All data were expressed as mean, with statistical significance determined at *p*-values < 0.05. Details on statistical parameters, including sample sizes and precision measures (e.g., *p*-values), are described in the figure legends or in the main text.

## Figures and Tables

**Figure 1 ijms-24-06433-f001:**
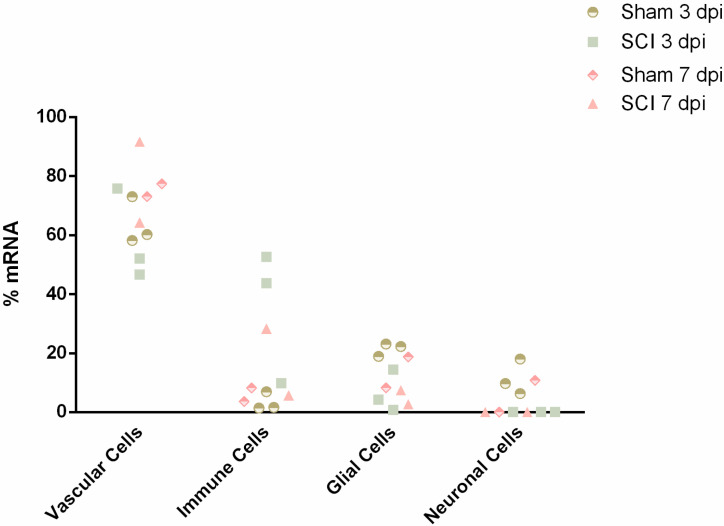
Digital cytometry analysis validates vascular cells as the overall most predominant cell type sorted by FACS at 3- and 7-days post-injury (n = 2–3). Single-cell gene expression data from spinal cord mouse tissue acquired from the Brain Mouse Atlas were used to build the cell signatures using 20 replicates from 50 cells classified with TaxonomyRank1 [19]. Next, cellular abundance was estimated with the bulk transcriptomic data using the subset of high-confidence genes. Cell signatures and cell type abundance were determined using CIBERSORTx [18].

**Figure 2 ijms-24-06433-f002:**
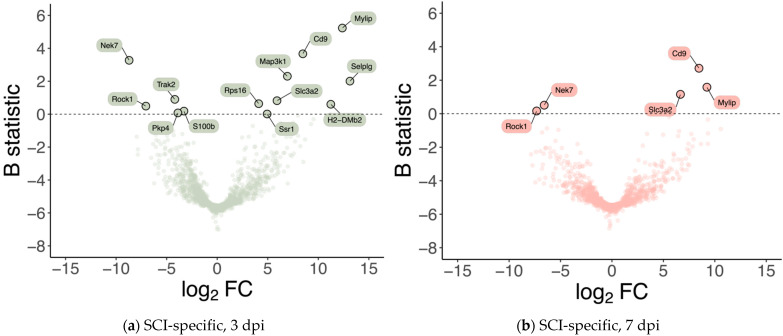
Spinal cord injury is associated with differences in the expression levels of a small subset of genes at 3- and 7-days post-injury. Volcano plots showing the effect size (in log_2_ fold-change; *X*-axis) and the significance (in B statistic, i.e., the log-odds ratio of differential expression; *Y*-axis) of differential gene expression specifically resulting from spinal cord injury (SCI) in (**a**) 3 and (**b**) 7 days post-injury (dpi) samples. Each point indicates one of the 1266 genes considered in the analysis.

**Figure 3 ijms-24-06433-f003:**
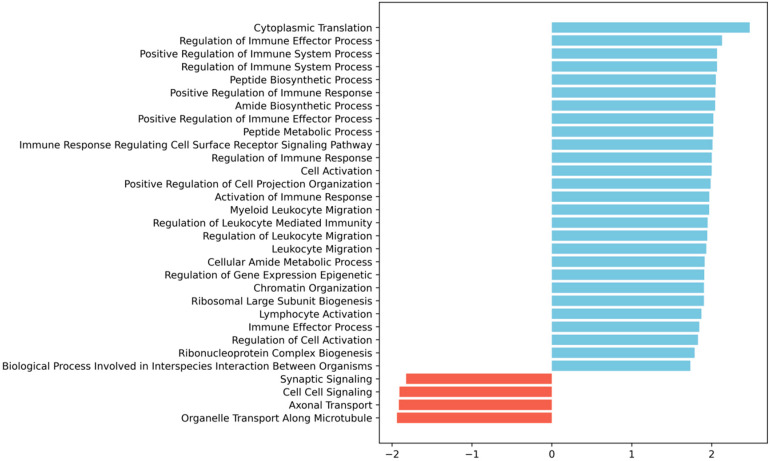
Gene Set Enrichment Analysis (GSEA) [20] of the 3 dpi SCI-specific transcriptomic alterations in vascular cells shows enrichment for immune- and leukocyte migration-related GO biological processes. The bar plot shows the significantly enriched terms ordered by normalized enrichment score (NES), blue bars for positive enrichment, and red bars for negative enrichment (i.e., depletion). GSEA was run as described in detail in the Materials and Methods section. We ran the GSEA pre-ranked method with empirical Bayes moderated t-statistic values against MsigDB’s gene ontology (GO) biological processes; terms were considered significantly enriched with an adjusted *p*-value < 0.05.

**Figure 4 ijms-24-06433-f004:**
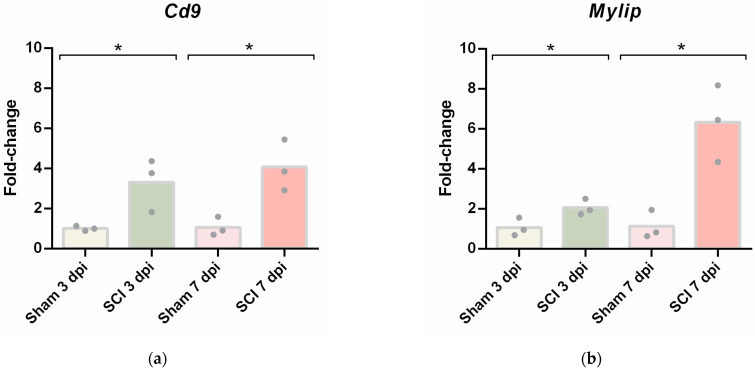
Spinal cord injury induces upregulation of *Cd9* and *Mylip* mRNA levels. (**a**) *Cd9* and (**b**) *Mylip* mRNA expression levels at 3- and 7-days post-injury (dpi) were evaluated by qPCR. For each cDNA sample, three technical replicates were included (n = 3). *Cd9* and *Mylip* mRNA levels were significantly increased with the injury at 3 and 7 dpi. Data are expressed in fold change towards the housekeeping gene PPIA, and bars represent the mean. * *p* < 0.05 versus sham, Student’s *t*-test.

**Figure 5 ijms-24-06433-f005:**
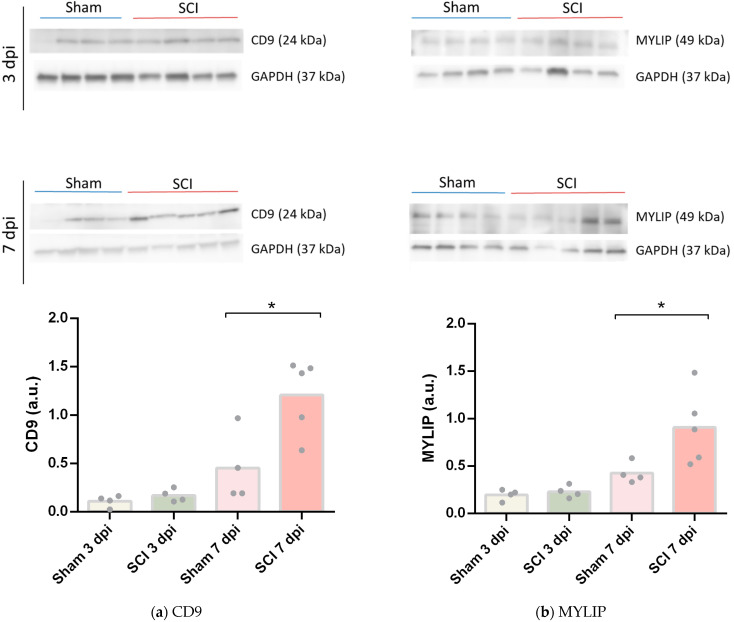
CD9 and MYLIP protein expression increase after spinal cord injury. Representative Western Blot images of CD9 (**a**) and MYLIP (**b**) with GAPDH as protein loading control at 3- and 7-days post-injury (dpi), each lane corresponds to an independent biological replicate. CD9 (**a**) and MYLIP (**b**) protein levels were quantified in sham and injured animals with a significant increase at 7 dpi for CD9 and MYLIP (n = 4–5). Data are expressed in relative intensity towards GAPDH, and bars represent the mean. * *p* < 0.05 versus sham, Student’s *t*-test.

**Figure 6 ijms-24-06433-f006:**
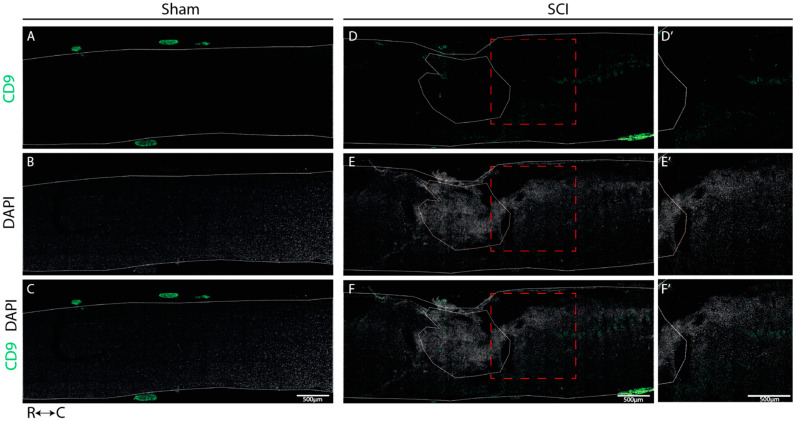
CD9 expression is injury-induced at the caudal side of the lesion. Representative images of sham (**A**–**C**) and injured spinal cords (**D**–**F**) demonstrate that CD9 (green) is only detected after injury, with more prominent expression in the caudal region of the lesion (**D’**–**F’**) in all animals analyzed (n = 3). DAPI in grey. White dashed lines: spinal cord delimitation and core of the lesion; Red insert: close-up of zone of interest; R: rostral; C: caudal. Scale bar: 500 µm; 20× amplification.

**Figure 7 ijms-24-06433-f007:**
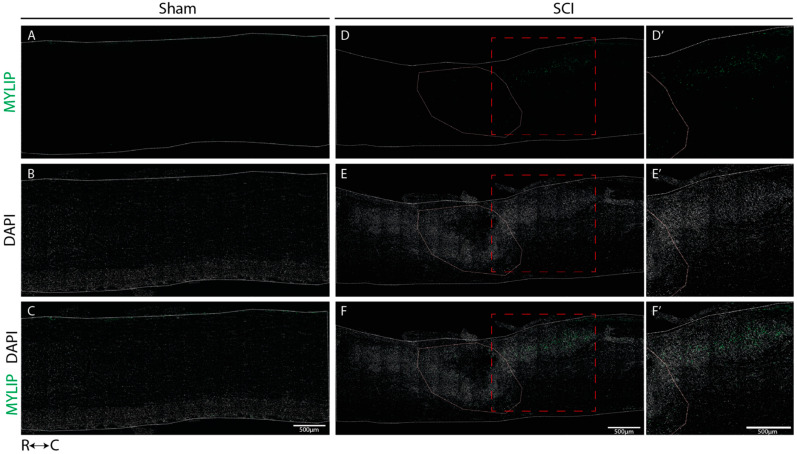
MYLIP is expressed only after injury at the caudal side of the lesion. Representative images of sham (**A**–**C**) and injured spinal cords (**D**–**F**) demonstrate that MYLIP (green) is injury-induced with more prominent expression in the caudal region of the lesion (**D’**–**F’**) in all animals analyzed (n = 3). DAPI in grey. White dashed lines: spinal cord delimitation and core of the lesion; Red insert: close-up of the zone of interest R: rostral; C: caudal. Scale bar: 500 µm; 20× amplification.

**Figure 8 ijms-24-06433-f008:**
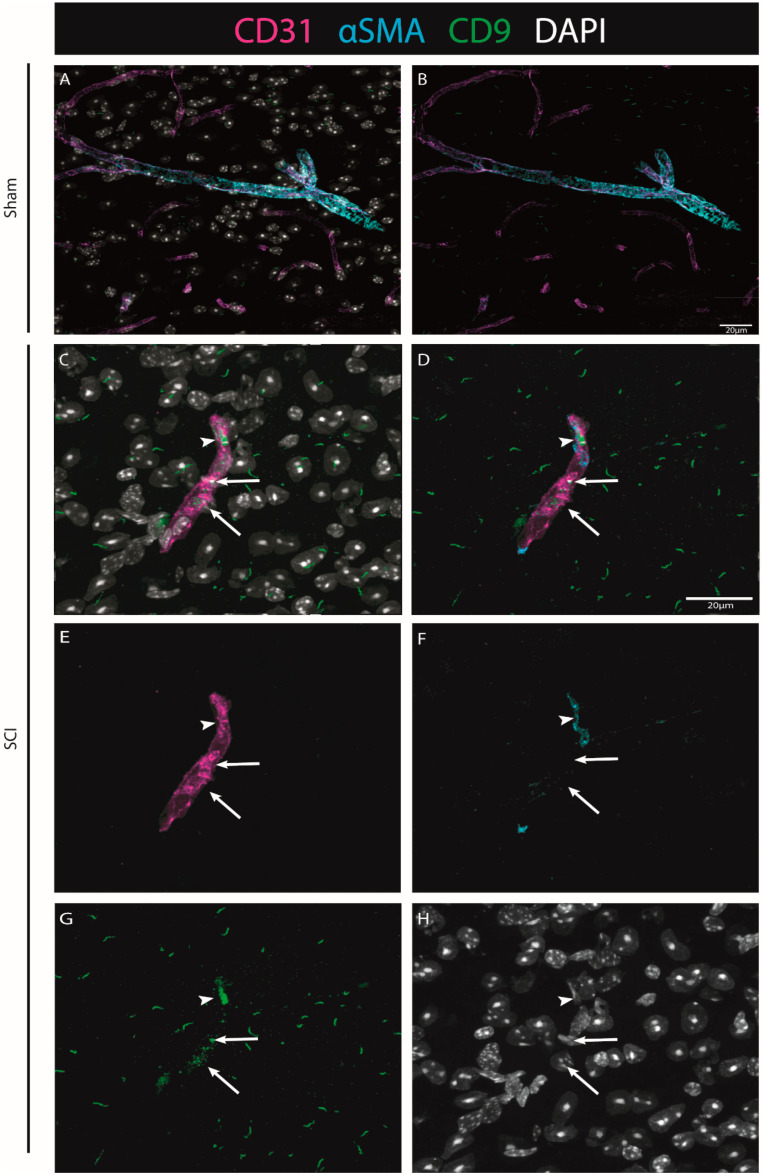
The injury-induced expression of CD9 is detected in close proximity to vascular and perivascular cells. Representative images of sham (**A**,**B**) and injured spinal cords (**C**–**H**) demonstrate that when CD9 (green) is induced after injury, it is detected in close proximity to both endothelial marker CD31 (magenta, arrows) and pericytic marker αSMA (cyan, arrowhead) at 7 dpi (n = 3). DAPI in grey. Scale bar: 20 µm; 40× amplification.

**Figure 9 ijms-24-06433-f009:**
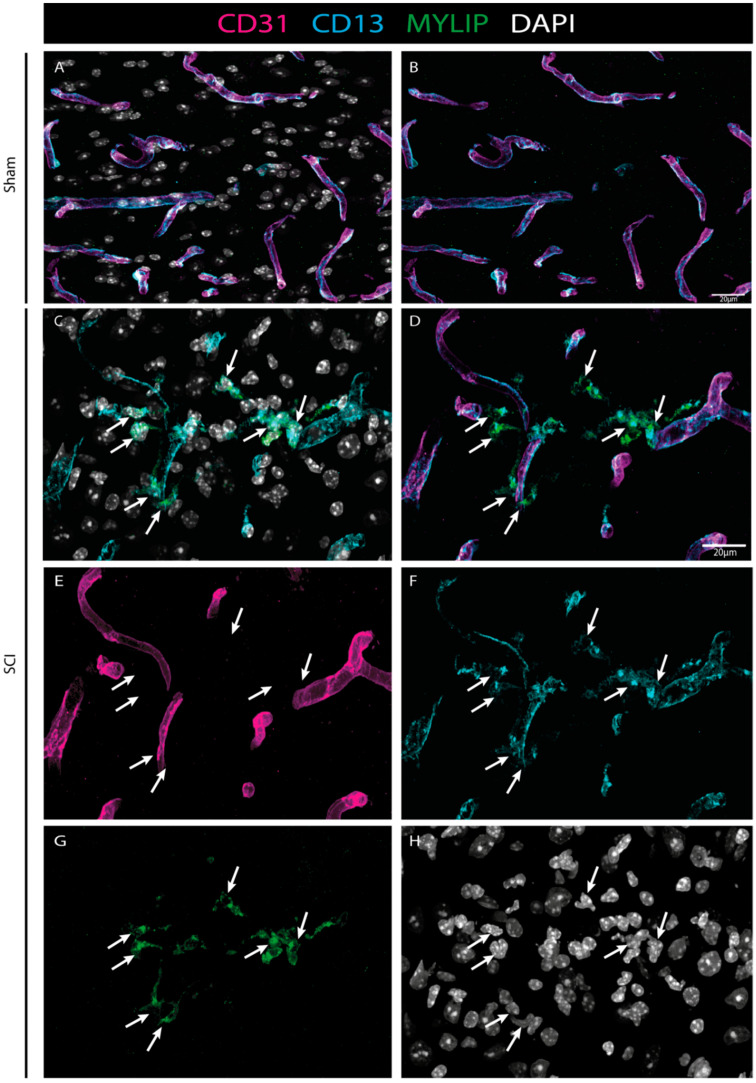
MYLIP expression is injury-induced in pericytes. Representative images of sham (**A**, **B**) and injured spinal cords (**C**–**H**) identify MYLIP (green) expression in close proximity with the pericytic marker CD13 (cyan, arrows), surrounding the blood vessels (CD31, magenta) at 7 dpi (n = 3). DAPI in grey. Scale bar: 20 µm; 40× amplification.

**Figure 10 ijms-24-06433-f010:**
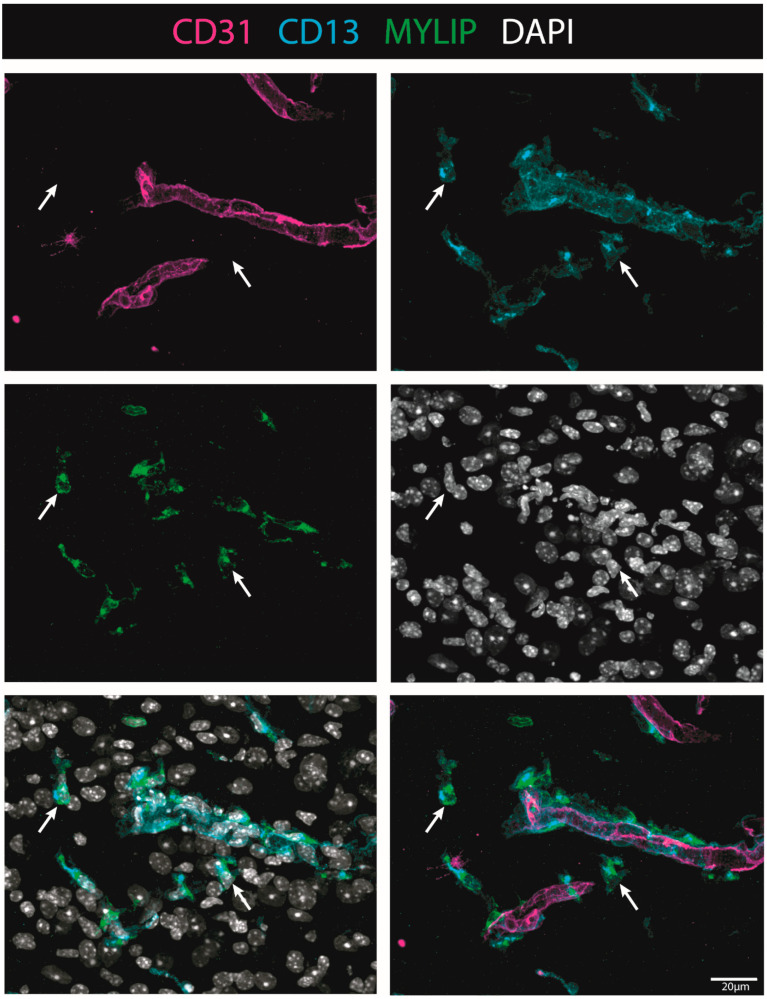
MYLIP is expressed in detached pericytes. Representative images of injured spinal cords demonstrate that MYLIP (green) is present within the pericytes (CD13, cyan, arrow) detached from blood vessels (CD31, magenta) at 7 dpi (n = 3). DAPI in grey. Scale bar: 20 µm; 40× amplification.

**Table 1 ijms-24-06433-t001:** Injury-specific differentially expressed genes at 3- and 7-days post-injury.

3 dpi	**Gene**	**logFC**	**B**	7 dpi	**Gene**	**logFC**	**B**
	*Mylip*	12.377	5.233		*Cd9*	8.458	2.715
	*Cd9*	8.469	3.666		*Mylip*	9.229	1.584
	*Nek7*	−8.711	3.270		*Scl3a2*	6.657	1.157
	*Map3k1*	6.939	2.303		*Nek7*	−6.585	0.510
	*Selplg*	13.130	2.006		*Rock1*	−7.321	0.163
	*Trak2*	−4.192	0.898				
	*Slc3a2*	5.915	0.819				
	*Rps16*	4.104	0.637				
	*H2-DMb2*	11.236	0.600				
	*Rock1*	−7.056	0.490				
	*S100b*	−3.289	0.185				
	*Pkp4*	−3.888	0.076				
	*Ssr1*	4.938	0.010				

dpi: days post-injury; logFC: log_2_ Fold-Change; B: B statistic.

## Data Availability

Not applicable.

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
