# Peer review of "Mouse Spinal Cord Vascular Transcriptome Analysis Identifies CD9 and MYLIP as Injury-Induced Players"

_ijms, 2023, doi:10.3390/ijms24076433_

Round 1
Reviewer 1 Report (Previous Reviewer 2)
In this manuscript authors analyze the transcriptome of spinal cord vascular cells in order to characterize the molecular basis of the response to injury in a mouse spinal cord injury model. They found CD9 and MYLIP expression to be injury-induced. The paper is easy to follow and written very well.
Author Response
We thank the reviewer for the comment and for the positive feedback on our manuscript.
Reviewer 2 Report (New Reviewer)
Considering my experience in spinal cord injury research I recommend this manuscript for publication in IJMS. However, I will refer to modifications if any question/enquiry "modifications required to seek approval" from the second reviewer, please.
I could see some yellow highlight areas in the manuscript text need clarifications by the authors.
Author Response
First of all we thank the reviewer for the positive review report.
In what regards the yellow highlight areas in the manuscript text it corresponds to changes and updates done on the original manuscript as requested by the editor for resubmission. This corresponds to small alterations as well as to the addition of new figures as requested previously.
This manuscript is a resubmission of an earlier submission. The following is a list of the peer review reports and author responses from that submission.
Round 1
Reviewer 1 Report
In this manuscript Martins et al highlight 2 genes which may be relevant to mechanisms underlying blood spinal cord barrier (BSCB) permeability following spinal cord injury. Utilising a clinically relevant contusion model in the mouse, they first FACS sort on CD31 (PECAM-1) for cells at 3 and 7 days post injury (dpi). Bulk RNA-seq is performed. Due to the lack of single-cell isolation, the genes are processed through CIBERSORT, a pipeline that estimates broad cell-type identity from bulk seq genes. Having assigned vascular cell-type, they run the genes through Gene Set Enrichment analysis (GSEA), and demonstrate enrichment for biological process relating to immune infiltration. They then highlight 2 genes, Cd9 and Mylip which appear consistently upregulated after injury. They perform western blot to attempt to show that these genes are also upregulated at a protein level, some of which is convincing. Immunostaining is then performed to again attempt to show upregulation after injury, though this is hard to see. Higher magnification microscopy and immunostaining of blood vessels/pericytes and the in-situ placement of cd9 and Mylip is then performed well, this is an interesting result and I can see promise for further study.
Whilst the paper is well written, and it is very commendable that the work tries to highlight novel processes for further study in a demanding clinically-relevent mouse model, the data is too preliminary for publishing in the International Journal of Molecular Sciences based on scope and impact.
I have included some comments which may aid the authors in preparing the manuscript, or some of the data within this manuscript, towards submission to a different journal:
I understand the requirement for CIBERSORT. It is a highly broad tool that is relatively non-sensitive. The basis of the paper on this lack-of-sensitivity is tricky, especially with the reduced n-number from technical issues with RNA-seq. But I do believe it is a valid starting point for the rest of the manuscript, particularly when looking at a fold-change level for the 2 genes with qPCR. I appreciate the current data may be a stepping-stone to acquisition of funding for these experiments, but there is no functional link between these genes and mechanisms of infiltration. Consider a transgenic mouse, or a gRNA against these genes, for example, to at least include loss of function.
Please make more clear in the text (not just the methods) that some groups in Figure 1 are n=2.
The western blots for mylip are not convincing, there is very little difference beyond technical artifact. For Cd9 it looks believable. For mylip, it might simply be that the 6mm tissue section is too large, especially if you think its spatially restricted to caudal to the lesion: I wouldn’t “force” this result with protein-level expression if the blots do not look clear.
The immunostaining in Figure 6 needs some work. It is very difficult to see what is going on here. Consider also staining for GFAP, for example, to demarcate the lesion rather than simply drawing it. Single channel images should usually be white to aid contrast, this is important here as it is very hard to see real staining. CD9 injured panel, there is what looks like “positive” staining outside the region drawn as the spinal cord borders. Its just not very convincing. Perhaps in-situ probes for RNA might help? Why is there so much staining in what looks like attached spinal roots? Consider including higher magnification insets for this staining, as it looks much more believable in Figure 7-9.
Mylip looks like it might be dura-related in the sham condition, although this could be a technical artifact, but it could be worth looking at this at higher magnification – many fibroblast-like cells express similar pericyte markers.
Figure 7 shows promise and is interesting data. Here there is a lot of Cd9 staining not associated with Cd31, is there a hypothesis for this? Consider staining for some immune-cell markers, as it looks to be in reasonable association with the DAPI.
Figure 8 and 9 - I would include sham immunostaining alongside this at high power, these are interesting data.
Reviewer 2 Report
In this manuscript authors assessed the transcriptome of spinal cord vascular cells in order to characterize the molecular basis of the response to injury in a mouse spinal cord injury model. They found CD9 and MYLIP expression to be injury-induced. The paper is easy to follow and written very well.
Minor points:
- line 336: please check if the strain is C57BL/6J or C57BL/6N.
- line 375: please correct Quiagen.
- line 577: please provide an approval number for study (protocol code XXX and date of approval).